# Effects of Restoration Strategies on the Ion Distribution and Transport Characteristics of *Medicago sativa* in Saline–Alkali Soil

**Baole Yu** [1,2], **Lingling Chen** [1,2,3,*] **and Taogetao Baoyin** [1,2,3]

1 Key Laboratory of Ecology and Resource Use of the Mongolian Plateau, Ministry of Education of China, School of Ecology and Environment, Inner Mongolia University, Hohhot 010021, China; baole.yu@gmail.com (B.Y.); bytgt@imu.edu.cn (T.B.)
2 Collaborative Innovation Center for Grassland Ecological Security, Ministry of Education of China, Inner Mongolia University, Hohhot 010021, China
3 Key Laboratory of Herbage and Endemic Crop Biology, Ministry of Education of China, School of Life Science, Inner Mongolia University, Hohhot 010021, China
* Correspondence: chenlingling@imu.edu.cn; Tel./Fax: +86-471-4992435

**Abstract:** Studying the distribution and transport dynamics of cations in plants is crucial for understanding their response mechanisms to saline–alkali stress conditions. However, our current understanding of how restoration measures affect cation distribution and transport in plants is surprisingly limited. To address this gap, we conducted a split-plot experiment using *Medicago sativa* L. cv. "Zhongmu No. 1" to investigate the combined effects of biological and chemical restoration measures—with bio-fertilizer as the primary zone and flue gas desulfurization (FGD) gypsum and with humic acid as the secondary zone—on soil properties, plant growth, and the content, distribution, and transport of cations in plants. The results revealed that bio-fertilizers exhibited positive effects on the plant growth, yield, and translocation of key ionic components to leaves. On the contrary, FGD gypsum with humic acid reduced the soil's pH level, exchangeable sodium percentage (ESP), and sodium adsorption ratio (SAR) while increasing the contents of $K^+$, $Ca^{2+}$, and $Mg^{2+}$ in the soil. The combination of bio-fertilizer, FGD gypsum, and humic acid increased the biomass and enhanced the translocation of $Mg^{2+}$ to leaves. The distribution and transport of $Mg^{2+}$ within the plant constituted pivotal elements for enhancing plant growth through restoration strategies. The application of bio-fertilizer, FGD gypsum, and humic acid reduced $Na^+$ transport in *M. sativa* by enhancing the selective absorption of beneficial ions in leaves and by facilitating the transport of $Ca^{2+}$ and $Mg^{2+}$ from stems to the leaves. This, in turn, increases the salt tolerance of plants and promotes their growth. Our results offer new insights into the interactions among measures, soil, and plants in saline–alkali land restoration, providing practical solutions for the restoration of saline–alkali soil.

**Keywords:** bio-fertilizer; flue gas desulfurization gypsum; humic acid; ion metabolism; saline–alkali land





## 1. Introduction

Environmental degradation, restricted agricultural development, and poverty resulting from increasing soil salinization have profoundly affected human society, particularly in the face of escalating global greenhouse gas emissions [1]. According to the statistical data published by the Agriculture Department of China, by the end of 2015, China had approximately $3.4 \times 10^7$ hm$^2$ saline–alkali soil in China [2]. Moreover, a notably high proportion of this land was moderate to severe saline–alkaline, with both the proportion and degree of saline–alkali soil significantly exceeding the global average [3]. In those saline–alkali soils, around $1.24 \times 10^7$ hm$^2$ can be potentially used for agricultural production after amelioration [2]. Effective restoration strategies and the judicious utilization of saline–alkali land is paramount for halting the salinization process, rectifying the issue



of insufficient arable land, and augmenting agricultural efficiency and livestock growth. However, previous research on the restoration of saline–alkali land has primarily been centered on plant yield and soil properties, often neglecting an evaluation of plant intrinsic response mechanisms.

Engineering, chemical, biological, and integrated approaches can rectify soil structure, enhance its physical and chemical attributes, bolster nutrient accumulation, and amplify crop yield, production performance, and plant nutrient quality [4–6]. Flue gas desulfurization (FGD) gypsum, a prominent by-product of power generation plants and a chemical amendment for saline–alkali land has shown promise. Recent studies indicate that FGD gypsum can lower soil pH and exchangeable sodium percentage (ESP), thereby expediting plant growth and improving salt tolerance [7,8]. Combining FGD gypsum with other soil-conditioning materials, such as organic fertilizer and humic acid, can decrease soil pH, electrical conductivity (EC), and $Na^+$ content while increasing soil porosity and $Mg^{2+}$ content, consequently boosting crop yield [8,9].

Targeted biological interventions, such as planting salt-tolerant species on saline–alkali land and applying microbial materials, are currently deemed the most effective and secure means of restoring saline–alkali lands [3,10]. Studies have demonstrated that biological measures not only diminish soil salinity during plant growth but also enhance the physicochemical properties of saline soils and facilitate the recovery of soil microbiomes [11,12].

Nevertheless, each of these engineering, chemical, and biological measures possesses its own set of drawbacks. For instance, engineering measures entail substantial human and material resources. The excessive application of chemical substances as soil conditioners may lead to secondary contamination of soil or plants or both. Biological measures often entail an unpredictable and protracted duration of persistent effects [13,14]. Thus, adopting an integrated and systematic approach, wherein various measures are applied judiciously and scientifically, is imperative to surmount these limitations. Using chemical and biological measures together can decrease the amount of chemical conditioning agents used. This approach maximizes overall improvement by leveraging combined effects. It is an efficient strategy to benefit from both methods while minimizing risks and addressing their weaknesses. In China, FGD gypsum and humic acid have been extensively researched and employed in saline land restoration, with growing interest in microbial-mediated restoration of saline land [15]. However, there has been relatively scant research on the combined use of these measures, especially in realistic field settings for saline–alkali grassland.

Plants employ a mechanism wherein they selectively absorb $K^+$ and $Ca^{2+}$ to elevate their internal $K^+/Na^+$ and $Ca^{2+}/Na^+$ ratios, maintaining heightened levels of $K^+$ and $Ca^{2+}$. This, in turn, mitigates the detrimental impact of $Na^+$ on the plant, thereby enhancing its salt resistance [5,16,17]. Despite various restoration measures potentially causing differential alterations in the ionic content and balance of plants, the underlying patterns and mechanisms remain poorly elucidated.

*Medicago sativa* L., known for its salinity tolerance, plays a pivotal role in enhancing the characteristics of saline–alkali soils [11,12,18]. Unsurprisingly, it finds widespread application in establishing artificial grasslands and fortifying the resilience of natural grasslands. As a result, it holds significant importance in both grassland ecosystem restoration and sustainable livestock development. However, the growth of *M. sativa* can be impeded under escalating salinity stress [12,19].

Recent research predominantly focuses on the impact of intervention measures on soil properties, plant growth, and crop quality. Rarely have there been reports on the distribution or transport of cations within plants under varying restoration measures, leaving the mechanisms behind most vegetation restoration largely unexplored. Hence, this study seeks to examine the effects of combined biological and chemical restoration measures on soil properties as well as the characteristics of ion distribution and transport in plants. This knowledge can be harnessed to enhance the salt tolerance of plants and expand the range of salt-tolerant species based on recommended practices. This, in turn, could

optimize the benefits of saline–alkali land restoration and utilization while mitigating the risk of secondary contamination.

With this objective in mind, this study aimed to assess the effects of FGD gypsum in conjunction with humic acid and bio-fertilizer on soil properties, plant growth, and the distribution and transport of ions in *M. sativa*. Additionally, it seeks to unravel the interplay between measures, soil, and plants. Three hypotheses were tested: (1) the applied restoration measures can variably reduce soil pH, exchangeable sodium percentage (ESP), and sodium adsorption ratio (SAR), thereby improving the soil environment; (2) the application of bio-fertilizer combined with FGD gypsum and humic acid amplifies the positive effects and is more conducive to plant growth; and (3) the restoration measures promote plant growth by facilitating the distribution of beneficial ions within plants and their subsequent transport to leaves.

## 2. Material and Methods

### 2.1. Experimental Site

The experimental saline–alkali soil was collected from Toketo County, Hohhot City, in the Inner Mongolia Autonomous Region, China. Toketo County is situated on the Tumocheon Plain at the southern foot of the Yin Mountains and on the northern bank of the upper and middle parts of the Yellow River ($111°2'30''\sim111°32'21''$ E, $40°5'55''\sim40°35'15''$ N). It falls within a temperate continental monsoon climate zone, with an average annual temperature of 9 °C and an active accumulated temperature ($\geq$10 °C) of 2961 °C. The average annual precipitation in this area is 316 mm, with 70% falling between July and September, and the average annual evaporation is 1938.2 mm.

### 2.2. Experimental Design

The experiment was conducted at the Shaerqin Experimental Station of the Institute of Grassland Research, Chinese Academy of Agricultural Sciences, from June to October 2021. The *M. sativa* cultivar employed in the experiment was "Zhongmu No. 1". Saline–alkali soil was collected from a depth of 40 cm in Manshui Village (Gucheng Town, Toketo County) over a 25 m$^2$ surface area (5 m × 5 m). After removing contaminants such as plant roots, the soil was thoroughly mixed and packed into plastic pots with a top diameter of 28 cm, a bottom diameter of 20 cm, and a height of 28 cm. Each pot was filled with 10 kg of soil, and all pots ($n = 40$) were buried in the experimental area; distances between pots were 50 cm. The basic properties of the soil and FGD gypsum used in the experiment are detailed in Table 1. The humic acid was manufactured by Dalian Jiucheng Products Co., Ltd. (Dalian, China), and the properties of the humic acid are detailed in Table 2; the bio-fertilizer was produced by Shandong Jinyao Biotechnology Co., Ltd. (Yangcheng, China), and the composition of the biofertilizers is *Bacillus subtilis*, with an effective live bacterial count of $\geq$200 × 10$^8$·g$^{-1}$.

The field experiment utilized a two-factor split-plot design, with the main plot receiving the applied bio-fertilizer (B) factor at two levels, which are 0 g·kg$^{-1}$ (B$_0$) and 6.0 g·kg$^{-1}$ (B$_6$). While the subplot received FGD gypsum with humic acid (D) at four levels, which are 0 g·kg$^{-1}$ (D$_0$), 7.5 + 0.75 g·kg$^{-1}$ (D$_{7.5}$), 15.0 + 1.5 g·kg$^{-1}$ (D$_{15}$), 30.0 + 3.0 g·kg$^{-1}$ (D$_{30}$). The dosage and ratio of FGD gypsum to humic acid (10:1) were determined based on relevant research. The dosage levels for each treatment are detailed in Table 3, with five replicates used for each. Each pot was sown with 20 fully developed *M. sativa* seeds at a distance of 3 cm, watered with groundwater twice or three times daily in small amounts during the first week post-sowing, and as needed afterward until seedling emergence. All treatments and their levels were managed uniformly in the field, and no additional fertilizer was applied during the experiment.

**Table 1.** Water-soluble ions in the initial soil and flue gas desulfurization gypsum were used in the present study.

| Index | Soil | FGD Gypsum |
|---|---|---|
| pH | 8.90 | $7.23 \pm 0.06$ |
| Water content/% | | 13 |
| EC/$\mu m \cdot cm^{-1}$ | 367.79 | |
| $Na^+$/$mg \cdot kg^{-1}$ | 54.2 | 574.0 |
| $K^+$/$mg \cdot kg^{-1}$ | 10.6 | 19.6 |
| $Ca^{2+}$/$mg \cdot kg^{-1}$ | 78.5 | 2731.0 |
| $Mg^{2+}$/$mg \cdot kg^{-1}$ | 24.8 | 55.0 |
| $Cl^-$/$mg \cdot kg^{-1}$ | 4.4 | 435.0 |
| $SO_4^{2-}$/$mg \cdot kg^{-1}$ | 15.6 | 7950.0 |
| $HCO_3^- + CO_3^{2-}$/$mg \cdot kg^{-1}$ | 363.0 | 126.0 |
| Exchangeable Na/cmol $(Na^+) \cdot kg^{-1}$ | 0.33 | 2.59 |

**Table 2.** Properties of humic acid used in the research.

| Index | Humic Acid |
|---|---|
| Humic acid content | 75% |
| Organic matter | 80% |
| pH | 4.46 |
| Exchangeable K/$g \cdot kg^{-1}$ | 50.56 |
| Exchangeable Na/$g \cdot kg^{-1}$ | 0.62 |
| Total N | 0.52% |
| $P_2O_5$ | 0.35% |
| $K_2O$ | 0.12% |

**Table 3.** The treatment scheme of different remediation measurements.

| Treatment | | Bio-Fertilizer /$g \cdot kg^{-1}$ | FGD Gypsum + Humic Acid /$g \cdot kg^{-1}$ |
|---|---|---|---|
| B0 | D0 | 0 | 0 |
| | D7.5 | 0 | 7.5 + 0.75 |
| | D15 | 0 | 15.0 + 1.5 |
| | D30 | 0 | 30.0 + 3.0 |
| B6 | D0 | 6.0 | 0 |
| | D7.5 | 6.0 | 7.5 + 0.75 |
| | D15 | 6.0 | 15.0 + 1.5 |
| | D30 | 6.0 | 30.0 + 3.0 |

Note: B0 and B6 indicate the treatments applied Bio-fertilizer with 0 $g \cdot kg^{-1}$ and 6.0 $g \cdot kg^{-1}$. D0, D7.5, D15, D30 indicate the treatments applied FGD gypsum + Humic acid with 0 $g \cdot kg^{-1}$, 7.5 + 0.75 $g \cdot kg^{-1}$, 15.0 + 1.5 $g \cdot kg^{-1}$, 30.0 + 3.0 $g \cdot kg^{-1}$, respectively.

### 2.3. Sampling and Measurements

After harvesting the plants in pots after seeding for 100 d, soil samples were collected. Initially, any debris was removed, followed by air-drying, grinding, and passing through a 1 mm aperture sieve. The soil was then shaken in a 5:1 water-to-soil ratio and allowed to stand before being filtered. Soil pH was measured using the potentiometry method, soil EC was determined using the electrode method, and soil salinity was derived using Pang et al.'s method [20]. For the quantification of soil water-soluble $HCO_3^-$ and $CO_3^{2-}$, the double-indicator titration method was employed; $Cl^-$ was assessed using the $AgNO_3$ titration method; $SO_4^{2-}$ was determined using the EDTA indirect titration method; $Ca^{2+}$ and $Mg^{2+}$ were measured using the EDTA complex titration method; and $Na^+$ and $K^+$ were quantified using the flame photometry method [21]. Cation exchange capacity (CEC) was

assessed via spectrophotometry [22], and exchangeable sodium (ES) was determined using flame photometry with $NH_4$ OAc-$NH_4$ OH [21].

### 2.3.1. Plant Growth and Biomass Variables

Before harvesting, five plants of similar size (i.e., growth) were selected from each pot to determine individual plant height, with the average value per pot calculated. Similarly, five plants of average growth were chosen to measure alfalfa's root diameter with vernier calipers, and the average value per pot was determined. All plants within the pots were harvested for the purpose of determining biomass. Subsequently, the harvested plants were segregated into leaves, stems, and roots. We measured the initial wet weights of leaves, stems, and roots and recorded the values. Subsequently, selected plant organs weighing between 200 and 300 g were chosen as samples for drying. Upon returning to the laboratory, these samples underwent initial heating at 105 °C for 30 min, followed by continuous drying at 65 °C until a consistent weight was obtained. They were then re-weighed (dry), and the biomass of each plant organ per replicate level was calculated [21].

### 2.3.2. Plant Ion Content

The dried samples of leaves, stems, and roots, processed according to the methodology outlined in the Section 2.3.1, need to be finely ground and sifted through a screen with an aperture size of 0.125 mm. Subsequently, 0.2 g of each sample was weighed, followed by the addition of 8 mL of $HNO_3$. The mixture was boiled using a graphite digestion apparatus until about 1 mL of liquid remained, after which 2 mL of $H_2O_2$ was introduced. The resulting digested solution was brought to a 50 mL volume with ultrapure water and passed through a 0.45-µm filter membrane. Finally, an inductively coupled plasma emission spectrometer (ICP-OES, Thermo Fisher Scientific, ICAP6300Duo, Waltham, MA, USA) was utilized to measure the contents of $Na^+$, $K^+$, $Ca^{2+}$, and $Mg^{2+}$.

### 2.4. Data Analysis

The sodium adsorption ratio (SAR) was calculated using the following formula [23]:

$$SAR = Na^+ / \sqrt{Ca^{2+} + Mg^{2+}}$$

where $Na^+$, $Ca^{2+}$, and $Mg^{2+}$ are the amounts of water-soluble $Na^+$, $Ca^{2+}$, and $Mg^{2+}$ in a soil sample.

The exchangeable sodium percentage (ESP) was calculated using the following formula [23]:

$$ESP\ (\%) = (ES/CEC) \times 100$$

The transport selectivity ratio (TS) indicates whether cations in transport are behaving synergistically or antagonistically. A greater TS value suggests that organ b is more effective in regulating $Na^+$ and facilitating the transport of Y ions to organ a. The equation for TS is as follows [12]:

$$TS\ (Y, Na^+) = organ\ a\ (Y/Na^+)/organ\ b\ (Y/Na^+),$$

where Y is the ion content; a and b are the leaf, stem, and/or root organ of the plant sample.

### 2.5. Statistical Analysis

To test the effects of both treatments on plant attributes (height, root length, root diameter, biomass, ion content, and ion transport indicators) and soil properties (pH, salinity, SAR, ESP, and amounts of water-soluble ions), a two-way analysis of variance (ANOVA) was conducted. This was followed by Tukey's HSD test (at $p < 0.05$), implemented in SPSS software (v23.0; SPSS Inc., Chicago, IL, USA). Each reported value represents the mean $\pm$ standard error of five individuals ($n = 5$). Pearson correlation tests were utilized to identify the relationships between plant and soil properties. Redundancy analysis (RDA) was employed to establish the relationships among soil properties, plant

growth indicators, and the biomass, ion content, and transport indicators of plants, using Canoco 5.0 software (Microcomputer Power B.V.; Wageningen; the Netherlands). Linear regression analysis was performed to quantify the relationship between soil environmental factors and the biomass of plants. For evaluating direct or indirect effects, structural equation modeling (SEM) was employed using AMOS software (v24.0; SPSS Inc., Chicago, IL, USA).

## 3. Results

### 3.1. pH, Salinity, ESP, SAR, and Water-Soluble Ion Content in the Soil

All treatments after application of FGD gypsum with humic acid led to a significant decrease in pH, ESP, and SAR ($p < 0.05$; see Figure 1a,c,d) while significantly increasing soil salinity ($p < 0.05$; see Figure 1b). The application of FGD gypsum with humic acid combined with biofertilizer showed significant interaction effects on pH, ESP, and soil salinity ($p < 0.05$; see Figure 1a–c). The application of FGD gypsum with humic acid combined with biofertilizer led to a significant interaction effect between soil salinity and ESP at D7.5 and a significant interaction effect between pH and ESP at D15.

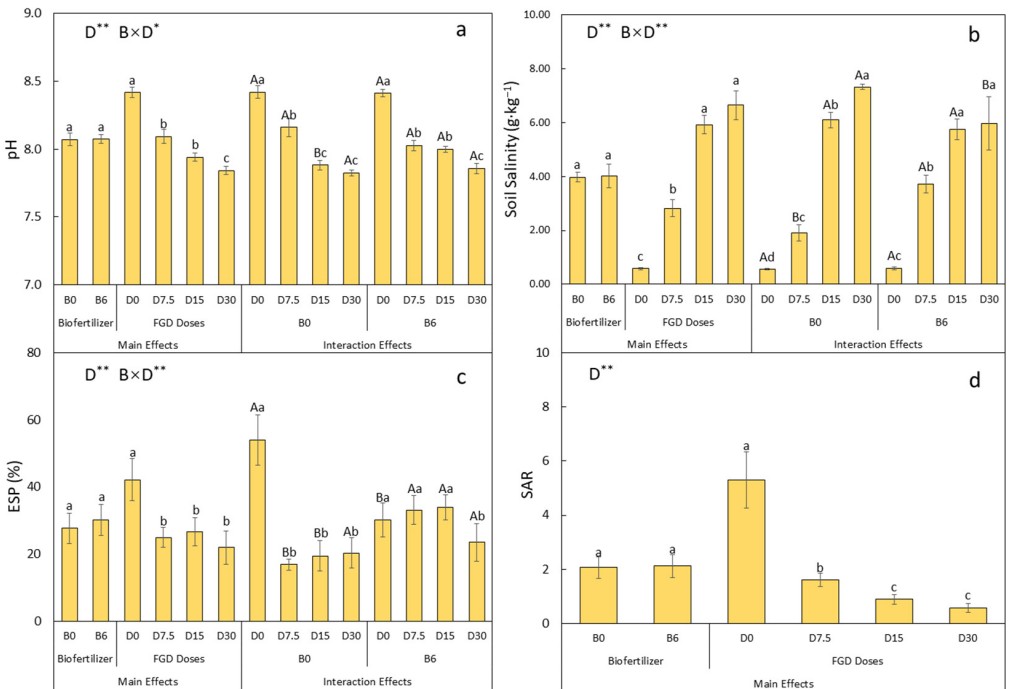

**Figure 1.** Effects of restoration strategies on soil pH, salinity, ESP, and SAR. (**a**) soil pH (**b**) soil salinity (**c**) ESP (**d**) SAR Note: B0 and B6 indicate the treatments applied Bio-fertilizer with 0 g·kg$^{-1}$ and 6.0 g·kg$^{-1}$. D0, D7.5, D15, D30 indicate the treatments applied FGD gypsum + Humic acid with 0 g·kg$^{-1}$, 7.5 + 0.75 g·kg$^{-1}$, 15.0 + 1.5 g·kg$^{-1}$, 30.0 + 3.0 g·kg$^{-1}$, respectively. Different lowercase letters represent the significant difference between levels of each treatment factor and between FGD doses in each level of biofertilizer, and different uppercase letters represent the significant difference between levels of biofertilizer in each level of FGD doses ($p < 0.05$). * indicates significant difference at 0.05 level. ** indicates significant difference at 0.01 level.

The water-soluble Na$^+$ content and ES were notably higher after the application of bio-fertilizer ($p < 0.05$; see Figure 2a,h). As for water-soluble K$^+$, Ca$^{2+}$, Mg$^{2+}$, Cl$^-$, SO$_4^{2-}$, HCO$_3^-$ + CO$_3^{2-}$, all their contents displayed significant variations under different applications of FGD gypsum with humic acid ($p < 0.05$; see Figure 2b–g). The application of FGD gypsum with humic acid combined with biofertilizer showed a significant interaction effect on water-soluble Na$^+$, K$^+$, Cl$^-$, SO$_4^{2-}$, HCO$_3^-$ + CO$_3^{2-}$ and ES ($p < 0.05$; see Figure 2a,b,e–h).

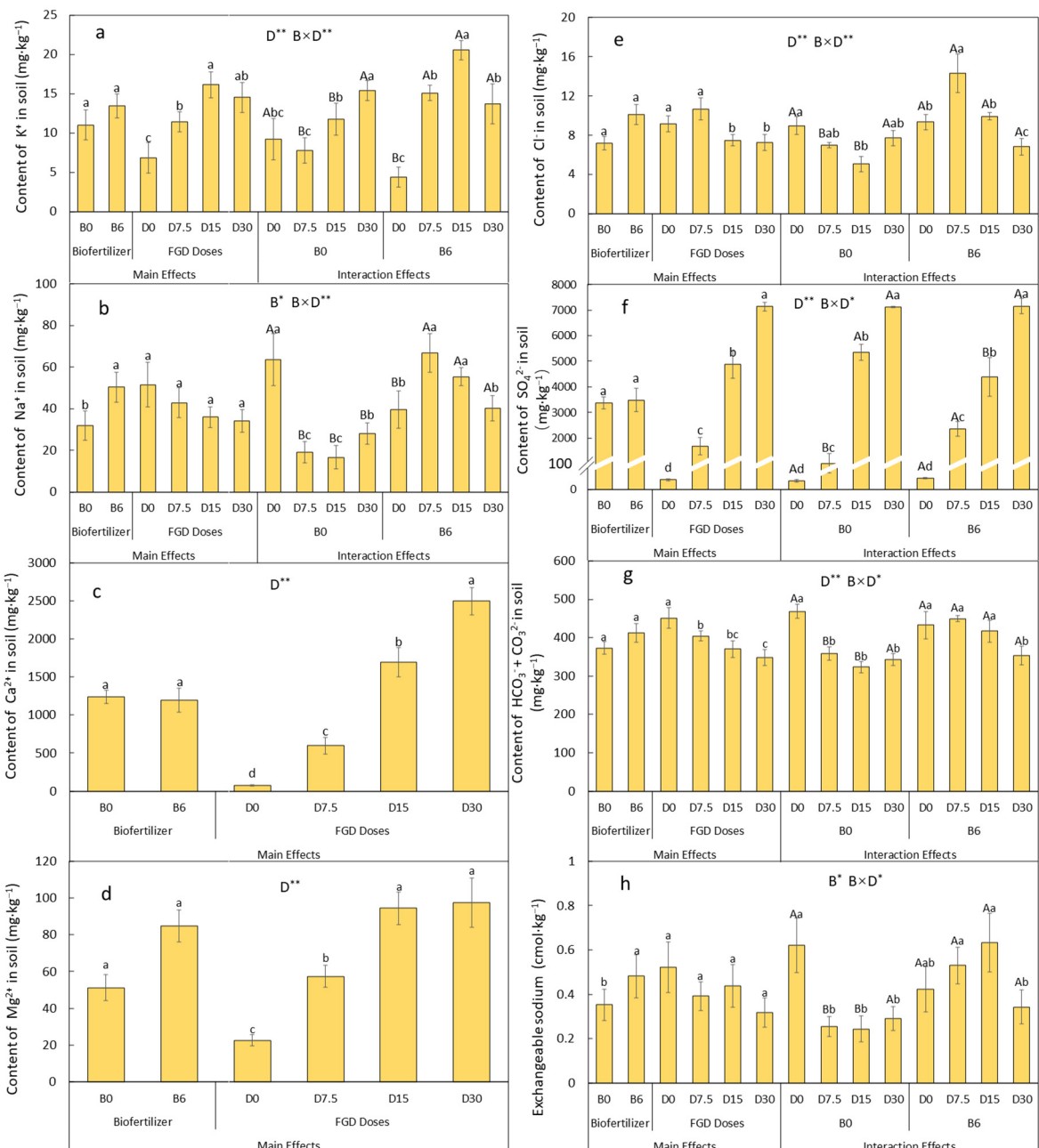

**Figure 2.** Effects of restoration strategies on the content of water-soluble ions. (**a**) contents of $K^+$ in soil (**b**) contents of $Na^+$ in soil (**c**) contents of $Ca^{2+}$ in soil (**d**) contents of $Mg^{2+}$ in soil (**e**) contents of $Cl^-$ in soil (**f**) contents of $SO_4^{2-}$ in soil (**g**) contents of $HCO_3^- + CO_3^{2-}$ in soil (**h**) exchangeable sodium (ES) in soil. Note: B0 and B6 indicate the treatments applied Bio-fertilizer with 0 g·kg$^{-1}$ and 6.0 g·kg$^{-1}$. D0, D7.5, D15, D30 indicate the treatments applied FGD gypsum + Humic acid with 0 g·kg$^{-1}$, 7.5 + 0.75 g·kg$^{-1}$, 15.0 + 1.5 g·kg$^{-1}$, 30.0 + 3.0 g·kg$^{-1}$, respectively. Different lowercase letters represent the significant difference between levels of each treatment factor and between FGD doses in each level of biofertilizer, and different uppercase letters represent the significant difference between levels of biofertilizer in each level of FGD doses ($p < 0.05$). * indicates significant difference at 0.05 level. ** indicates significant difference at 0.01 level.

### 3.2. Plant Growth Indicators and Biomass

The application of bio-fertilizers significantly enhanced plant height, root length, and root diameter ($p < 0.05$; see Figure 3), along with the biomass of each organ, as well as the overall biomass, of *M. sativa* plants ($p < 0.05$; see Figure 4). As for the biomass of each organ

and total biomass displayed significant variations under different applications of FGD gypsum with humic acid, D15 was highest ($p < 0.05$; see Figure 4). The application of FGD gypsum with humic acid combined with biofertilizer showed a significant interaction effect on stem biomass, root biomass, and total biomass. The combination led to a significant increase in the stem biomass, root biomass, and total biomass at D7.5 and D15 ($p < 0.05$; see Figure 4b–d).

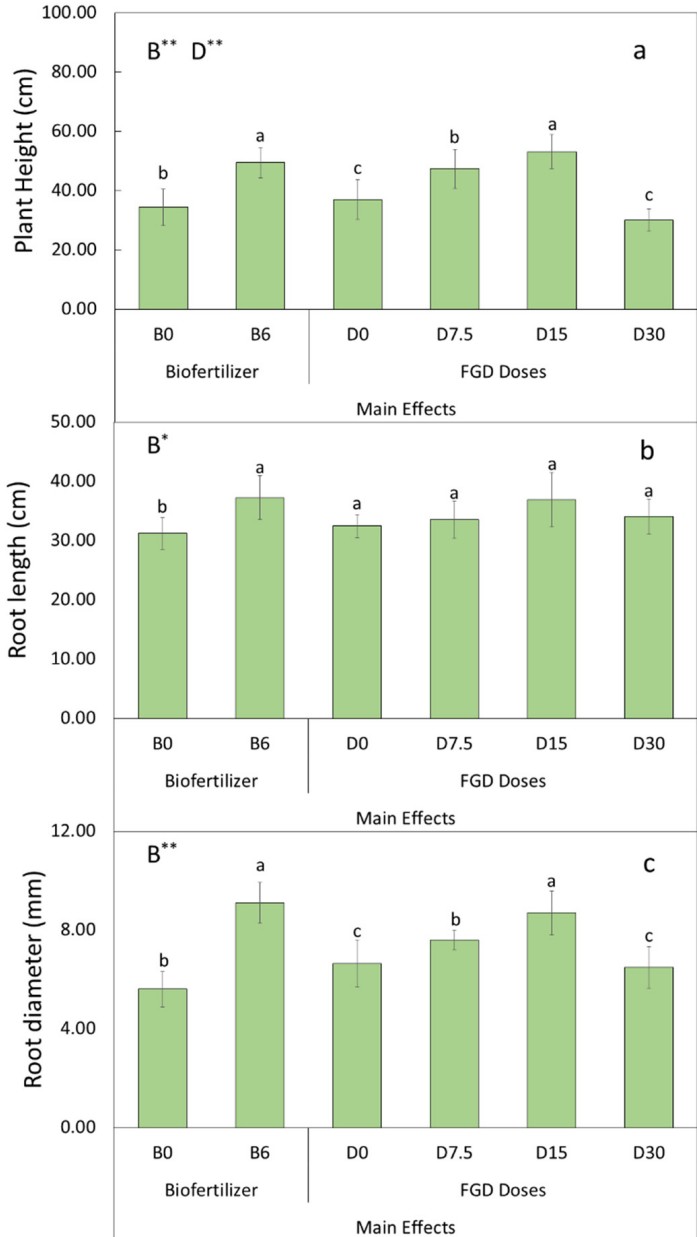

**Figure 3.** Effects of restoration strategies on the height, root length, and root diameter of *M. sativa*. (**a**) height (**b**) root length (**c**) root diameter Note: B0 and B6 indicate the treatments applied Biofertilizer with 0 g·kg$^{-1}$ and 6.0 g·kg$^{-1}$. D0, D7.5, D15, D30 indicate the treatments applied FGD gypsum + Humic acid with 0 g·kg$^{-1}$, 7.5 + 0.75 g·kg$^{-1}$, 15.0 + 1.5 g·kg$^{-1}$, 30.0 + 3.0 g·kg$^{-1}$, respectively. Different lowercase letters represent the significant difference between levels of each treatment factor and between FGD doses in each level of biofertilizer ($p < 0.05$). * indicates significant difference at 0.05 level. ** indicates significant difference at 0.01 level.

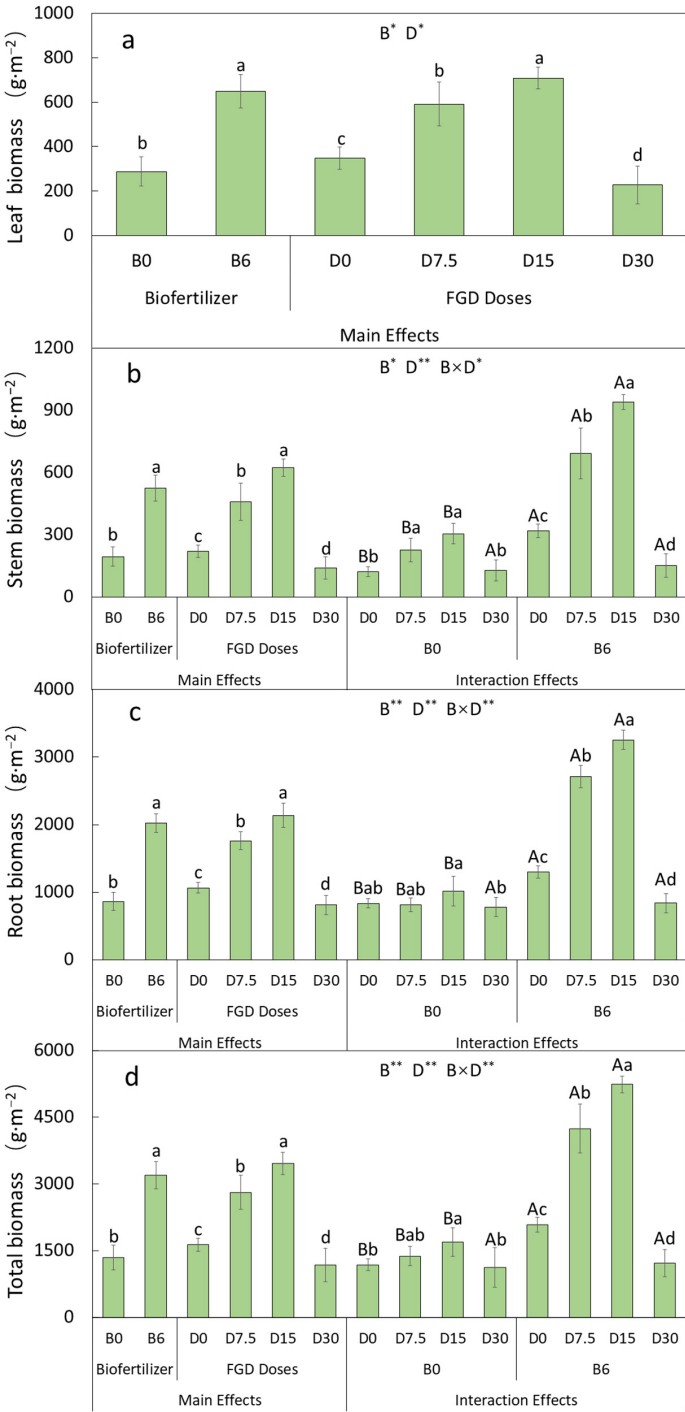

**Figure 4.** Effects of restoration strategies on the aboveground biomass and belowground biomass of the *M. sativa*. (**a**) leaf biomass (**b**) stem biomass (**c**) root biomass (**d**) total biomass Note: B0 and B6 indicate the treatments applied Bio-fertilizer with 0 g·kg$^{-1}$ and 6.0 g·kg$^{-1}$. D0, D7.5, D15, D30 indicate the treatments applied FGD gypsum + Humic acid with 0 g·kg$^{-1}$, 7.5 + 0.75 g·kg$^{-1}$, 15.0 + 1.5 g·kg$^{-1}$, 30.0 + 3.0 g·kg$^{-1}$, respectively. Different color lines link the treatments to indicate different organs of *M. sativa*, black line indicates total biomass, deep green line indicates stem, light green indicates leaf, and yellow indicates root. Different lowercase letters represent the significant difference between levels of each treatment factor and between FGD doses in each level of biofertilizer, and different uppercase letters represent the significant difference between levels of biofertilizer in each level of FGD doses ($p < 0.05$). * indicates significant difference at 0.05 level. ** indicates significant difference at 0.01 level.

### 3.3. Ionic Effects in M. sativa

### 3.3.1. Concentrations of Na$^+$, K$^+$, Ca$^{2+}$, and Mg$^{2+}$

Leaf K$^+$ levels were notably affected by the doses of FGD gypsum with humic acid. Additionally, the application of bio-fertilizer significantly increased leaf Mg$^{2+}$ while decreasing stem K$^+$ (see Figure 5). The application of FGD gypsum with humic acid combined with biofertilizer showed a significant interaction effect on the content of K$^+$ in the leaf ($p < 0.05$; see Figure 5).

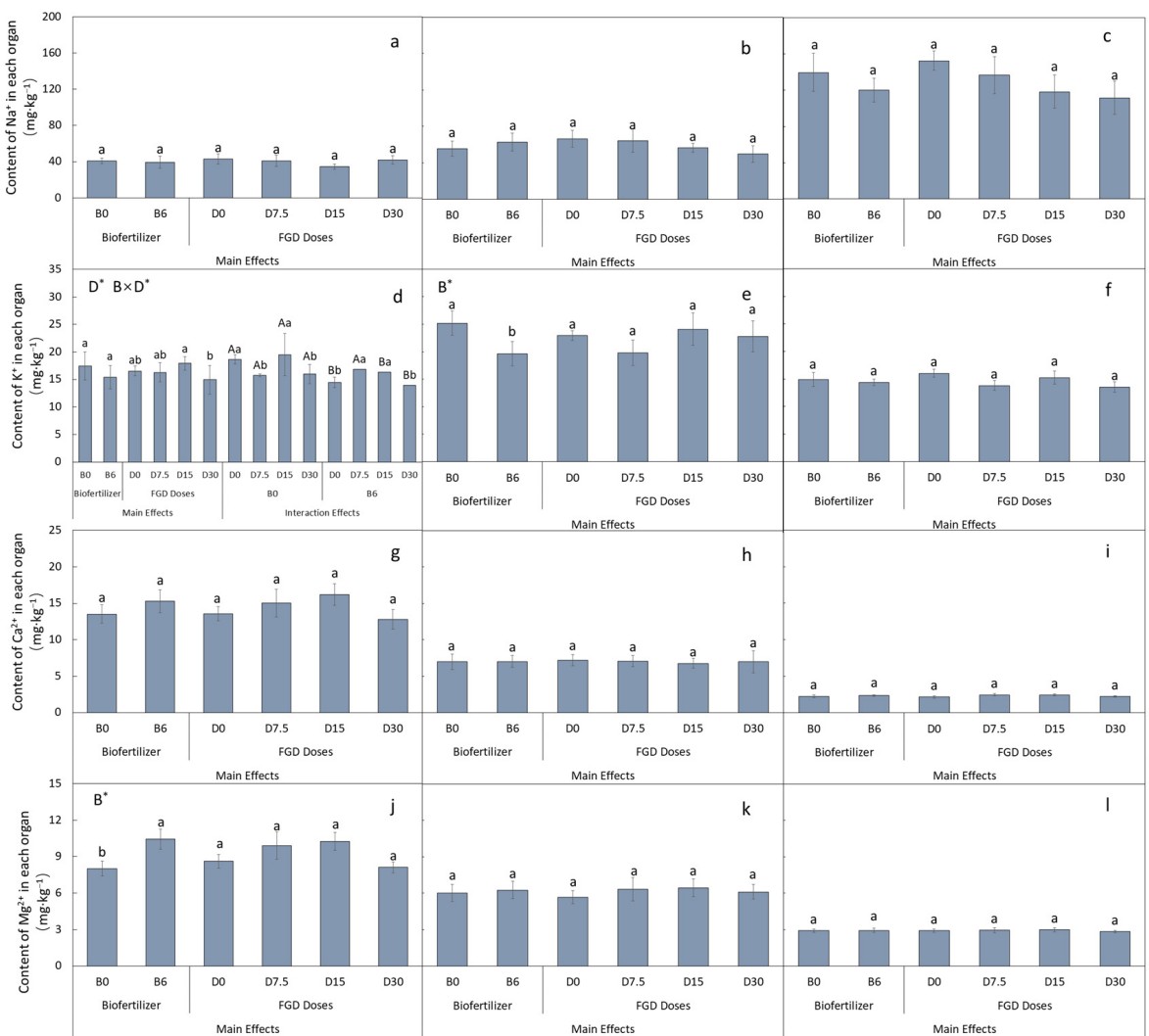

**Figure 5.** Effects of restoration strategies on the content of ions in each organ of *M. sativa*. (**a**) content of Na$^+$ in leaf (**b**) content of Na$^+$ in stem (**c**) content of Na$^+$ in root (**d**) content of K$^+$ in leaf (**e**) content of K$^+$ in stem (**f**) content of K$^+$ in root (**g**) content of Ca$^{2+}$ in leaf (**h**) content of Ca$^{2+}$ in stem (**i**) content of Ca$^{2+}$ in root (**j**) content of Mg$^{2+}$ in leaf (**k**) content of Mg$^{2+}$ in stem (**l**) content of Mg$^{2+}$ in root. Note: B0 and B6 indicate the treatments applied Bio-fertilizer with 0 g·kg$^{-1}$ and 6.0 g·kg$^{-1}$. D0, D7.5, D15, D30 indicate the treatments applied FGD gypsum + humic acid with 0 g·kg$^{-1}$, 7.5 + 0.75 g·kg$^{-1}$, 15.0 + 1.5 g·kg$^{-1}$, 30.0 + 3.0 g·kg$^{-1}$, respectively. Different lowercase letters represent the significant difference between levels of each treatment factor and between FGD doses in each level of biofertilizer, and different uppercase letters represent the significant difference between levels of biofertilizer in each level of FGD doses ($p < 0.05$). * indicates significant difference at 0.05 level.

### 3.3.2. Ratios of $K^+/Na^+$, $Ca^{2+}/Na^+$, $Mg^{2+}/Na^+$ in Plant Organs

Both leaf $K^+/Na^+$ and $Mg^{2+}/Na^+$ ratios saw significant increases under the medium dose of FGD gypsum with humic acid. However, the application of bio-fertilizer significantly increased the leaf $Ca^{2+}/Na^+$ and $Mg^{2+}/Na^+$ ratios while lowering the stem $K^+/Na^+$ ratio (see Figure 6).

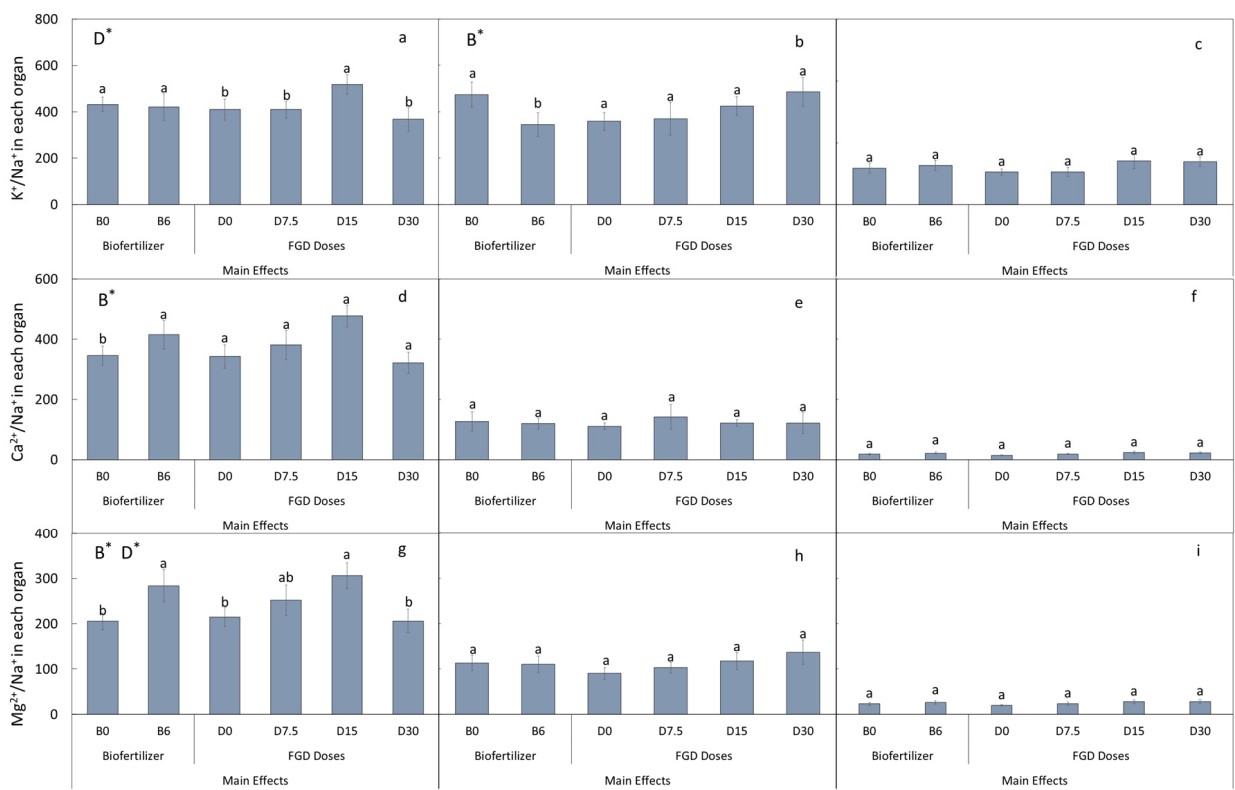

**Figure 6.** Effects of restoration strategies on the $K^+/Na^+$, $Ca^{2+}/Na^+$, $Mg^{2+}/Na^+$ in each organ of *M. sativa* (**a**) $K^+/Na^+$ in leaf (**b**) $K^+/Na^+$ in stem (**c**) $K^+/Na^+$ in root (**d**) $Ca^{2+}/Na^+$ in leaf (**e**) $Ca^{2+}/Na^+$ in stem (**f**) $Ca^{2+}/Na^+$ in root (**g**) $Mg^{2+}/Na^+$ in leaf (**h**) $Mg^{2+}/Na^+$ in stem (**i**) $Mg^{2+}/Na^+$ in root. Note: B0 and B6 indicate the treatments applied Bio-fertilizer with $0 \text{ g·kg}^{-1}$ and $6.0 \text{ g·kg}^{-1}$. D0, D7.5, D15, D30 indicate the treatments applied FGD gypsum + humic acid with $0 \text{ g·kg}^{-1}$, $7.5 + 0.75 \text{ g·kg}^{-1}$, $15.0 + 1.5 \text{ g·kg}^{-1}$, $30.0 + 3.0 \text{ g·kg}^{-1}$, respectively. Different lowercase letters represent the significant difference between levels of each treatment factor and between FGD doses in each level of biofertilizer ($p < 0.05$). * indicates significant difference at 0.05 level.

### 3.3.3. Cation TS Ratio of *M. sativa*

The stem-to-leaf selectivity ratio of $K^+$ showed significant variation under subplot treatments, with the highest ratio observed at D7.5. Furthermore, the stem-to-leaf selectivity ratio of $Ca^{2+}$ was significantly elevated by the application of bio-fertilizers. As for $Mg^{2+}$, its ratio was notably affected by both the primary and secondary zones. In particular, the application of bio-fertilizer substantially increased the ratio for $Mg^{2+}$ (see Figure 7). The application of FGD gypsum with humic acid combined with biofertilizer showed a significant interaction effect on the stem-to-leaf transport selectivity ratio of $Mg^{2+}$ ($p < 0.05$; see Figure 7).

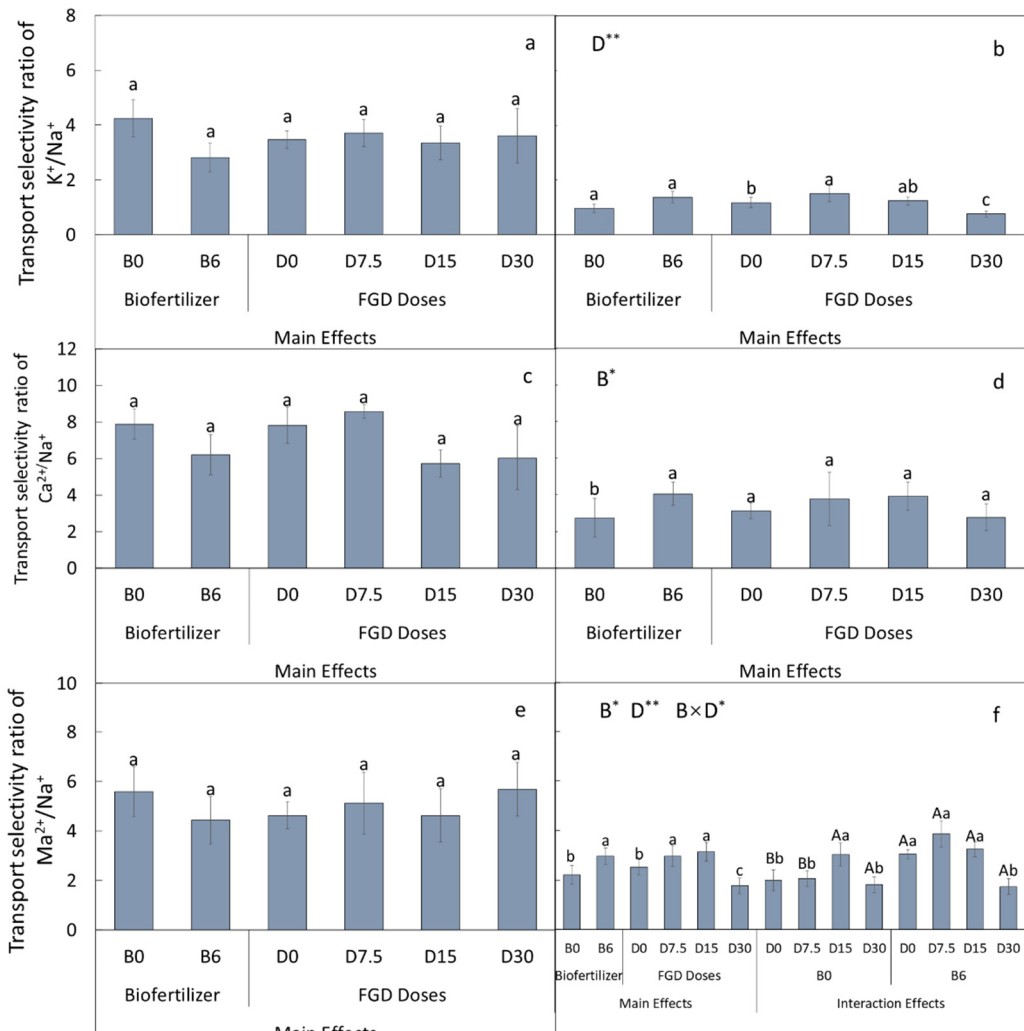

**Figure 7.** Effects of restoration strategies on cation transport selectivity ratio of ions in *Medicago sativa*. (**a**) root-to-stem $K^+$ transport selectivity ratio (**b**) stem-to-leaf $K^+$ transport selectivity ratio (**c**) root-to-stem $Ca^{2+}$ transport selectivity ratio (**d**) stem-to-leaf $Ca^{2+}$ transport selectivity ratio (**e**) root-to-stem $Mg^{2+}$ transport selectivity ratio (**f**) stem-to-leaf $Mg^{2+}$ transport selectivity ratio. Note: B0 and B6 indicate the treatments applied Bio-fertilizer with 0 $g \cdot kg^{-1}$ and 6.0 $g \cdot kg^{-1}$. D0, D7.5, D15, D30 indicate the treatments applied FGD gypsum + humic acid with 0 $g \cdot kg^{-1}$, 7.5 + 0.75 $g \cdot kg^{-1}$, 15.0 + 1.5 $g \cdot kg^{-1}$, 30.0 + 3.0 $g \cdot kg^{-1}$, respectively. Different lowercase letters represent the significant difference between levels of each treatment factor and between FGD doses in each level of biofertilizer, and different uppercase letters represent the significant difference between levels of biofertilizer in each level of FGD doses ($p < 0.05$). * indicates significant difference at 0.05 level. ** indicates significant difference at 0.01 level.

### 3.4. Correlation Analysis

According to the Pearson correlation coefficients, the biomass of *M. sativa* showed significant correlations with soil $Na^+$, $Cl^-$, $HCO_3^-$, and ESP. Plant height and biomass exhibited positive correlations with leaf $Ca^{2+}$ and $Mg^{2+}$ contents, leaf $Ca^{2+}/Na^+$ and $Mg^{2+}/Na^+$ ratios, and the stem-to-leaf selectivity ratio of $K^+$ as well as $Mg^{2+}$. The biomass values of different organs were significantly and negatively correlated with the root-to-stem selectivity ratios of $K^+$, $Ca^{2+}$, and $Mg^{2+}$ (see Figure 8).

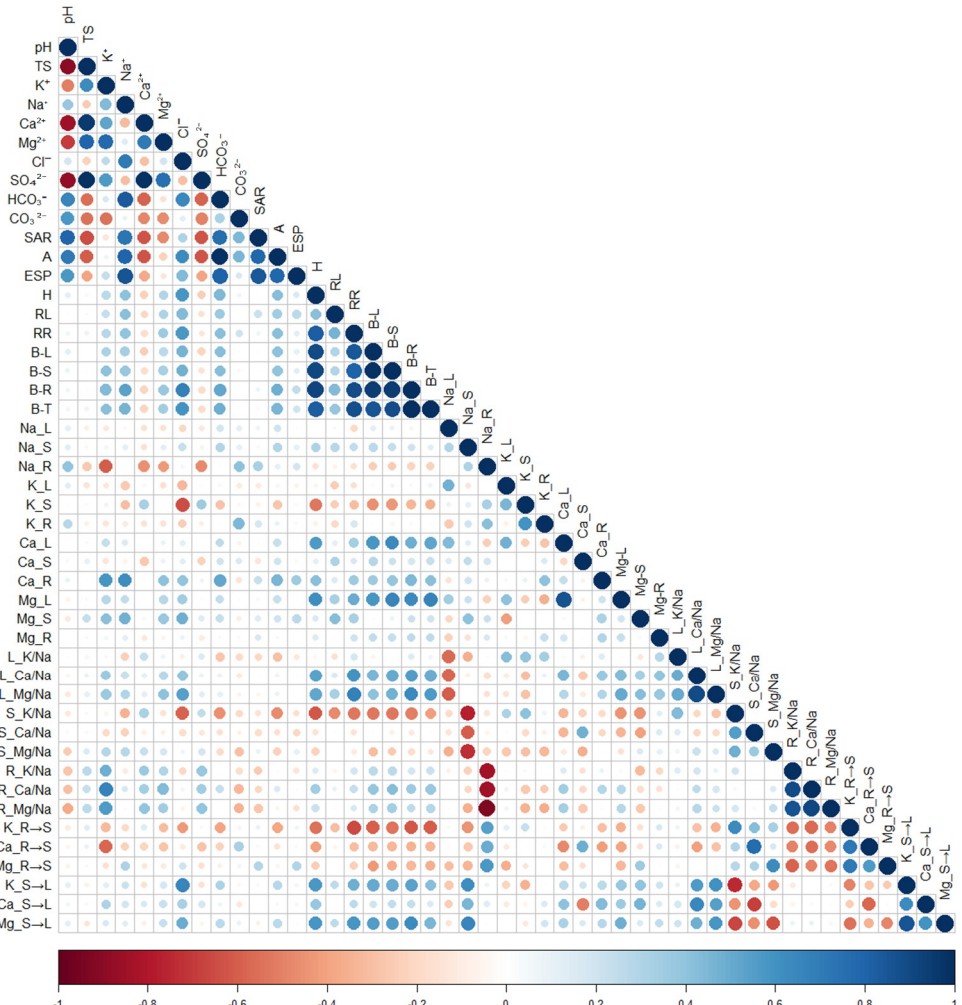

**Figure 8.** Correlation analysis of plant growth, ion partitioning, transport, and soil environmental factors under different remediation measures. TS: soil solidity, B-L, B-S, B-R, B-T: biomass of leaf, stem, and root, respectively. Na_L, K_L, Ca_L, Mg-L were content of $Na^+$, $K^+$, $Ca^{2+}$, $Mg^{2+}$ in leaf; Na_S, K_S, Ca_S, Mg-S: content of $Na^+$, $K^+$, $Ca^{2+}$, $Mg^{2+}$ in stem, Na_R, K_R, Ca_R, Mg_R: content of $Na^+$, $K^+$, $Ca^{2+}$, $Mg^{2+}$ in root, respectively. L_K/Na, L_Ca/Na, L_Mg/Na were $K^+/Na^+$, $Ca^+/Na^+$, $Mg^+/Na^+$ of leaf, S_K/Na, S_Ca/Na, S_Mg/Na were $K^+/Na^+$, $Ca^+/Na^+$, $Mg^+/Na^+$ of stem, R_K/Na, R_Ca/Na, R_Mg/Na were $K^+/Na^+$, $Ca^+/Na^+$, $Mg^+/Na^+$ of root, respectively. K_R→S, Ca_R→S, Mg_R→S: transport selectivity ratio of $K^+$, $Ca^{2+}$, $Mg^{2+}$ from root to stem, K_S→L, Ca_S→L, Mg_S→L: transport selectivity ratio of $K^+$, $Ca^{2+}$, $Mg^{2+}$ from stem to leaf.

The RDA demonstrated the significant influence of the soil environment on plant growth and biomass. Both responses were highly correlated with ESP. While soil pH positively influenced growth and biomass, soil salinity (TS) had a negative effect. However, neither pH nor TS emerged as crucial environmental factors. The soil contents of $Cl^-$ and $Mg^{2+}$ each exerted a significant positive effect on the growth and biomass of *M. sativa* ($p < 0.05$). Additionally, the length, diameter, and biomass of roots each showed positive correlations with soil $K^+$, $Mg^{2+}$, and $HCO_3^-$ (see Figure 8). The RDA also indicated a strong positive correlation between the ion content of roots and the corresponding ion content of soil. Notably, the $Na^+$ content in organs was highly correlated with that in the soil. Furthermore, the $Mg^{2+}$ content in soil had a robust positive effect on the ion content and transport dynamics in *M. sativa*, with a significant positive correlation between the $Mg^{2+}$ content in roots or stems and that in the soil. The selective transport ratios of $K^+$, $Ca^{2+}$, and $Mg^{2+}$ from stems to leaves, as well as that of $Mg^{2+}$ from roots to stems, were positively correlated with the amounts of $Na^+$, $K^+$, and $Mg^{2+}$ in the soil, whereas those of

$K^+$ and $Ca^{2+}$ from roots to stems were negatively correlated with the amounts of $Na^+$, $K^+$, and $Mg^{2+}$ in the soil (see Figure 9).

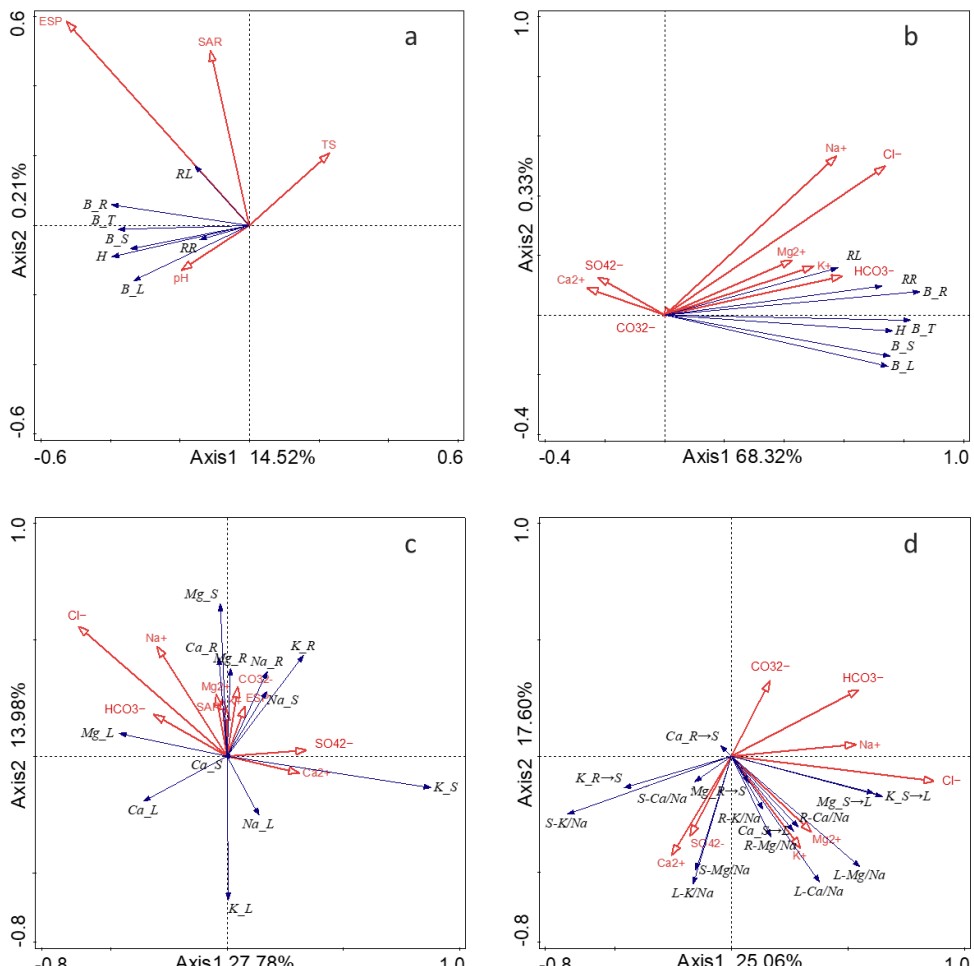

**Figure 9.** RDA analysis of plant growth, ion partitioning, transport, and soil environmental factors under different remediation measures (**a**) effect of soil salinity- alkalinity properties on plant growth and yield (**b**) effect of soil water-soluble cations on plant growth and yield (**c**) effect of soil water-soluble cations on contents of ions in plant (**d**) effect of soil water-soluble cations on transportation of ions in plant. TS: soil solidity, B-L, B-S, B-R, B-T: biomass of leaf, stem, and root, respectively. Na_L, K_L, Ca_L, Mg-L were content of $Na^+$, $K^+$, $Ca^{2+}$, $Mg^{2+}$ in leaf; Na_S, K_S, Ca_S, Mg-S: content of $Na^+$, $K^+$, $Ca^{2+}$, $Mg^{2+}$ in stem, Na_R, K_R, Ca_R, Mg_R: content of $Na^+$, $K^+$, $Ca^{2+}$, $Mg^{2+}$ in root, respectively. L_K/Na, L_Ca/Na, L_Mg/Na were $K^+/Na^+$, $Ca^+/Na^+$, $Mg^+/Na^+$ of leaf, S_K/Na, S_Ca/Na, S_Mg/Na were $K^+/Na^+$, $Ca^+/Na^+$, $Mg^+/Na^+$ of stem, R_K/Na, R_Ca/Na, R_Mg/Na were $K^+/Na^+$, $Ca^+/Na^+$, $Mg^+/Na^+$ of root, respectively. K_R→S, Ca_R→S, Mg_R→S: transport selectivity ratio of $K^+$, $Ca^{2+}$, $Mg^{2+}$ from root to stem, K_S→L, Ca_S→L, Mg_S→L: transport selectivity ratio of $K^+$, $Ca^{2+}$, $Mg^{2+}$ from stem to leaf.

In the SEM conducted in this study, the soil ion latent variables consisted of $K^+$, $Na^+$, $Cl^-$, and $HCO_3^-$, while the plant growth latent variables encompassed plant height, root diameter, leaf, stem, and root biomass. The SEM results illuminated that the combination of FGD gypsum with humic acid and bio-fertilizer significantly influenced plant growth. Furthermore, FGD gypsum with humic acid exhibited a greater impact on soil properties, while bio-fertilizer had a more pronounced effect on cation partitioning and transport in *M. sativa* plants (see Figure 10).

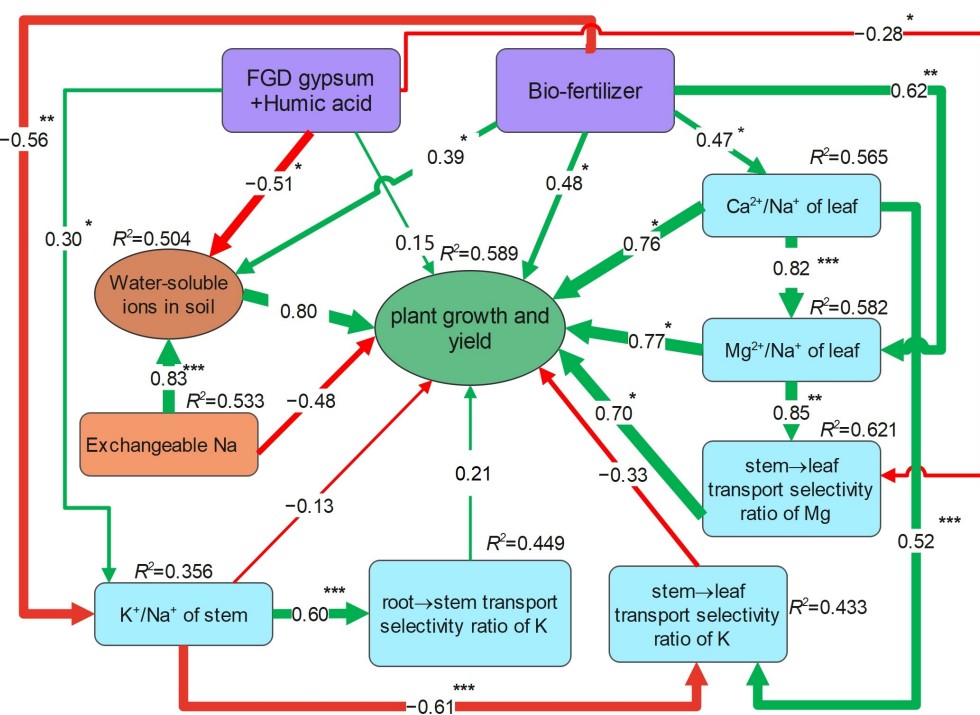

$x^2$=10.87, GFI=0.923, CFI=0.857, RMR=0.053, RMSEA=0.062

**Figure 10.** Structural equation modeling of the relationship among soil properties, plant growth, and ion transport characteristics under different remediation measures. The green arrow represents positive correlation, the red arrow represents negative correlation, the number on the arrow is the normalized path coefficient, and the width of the arrow indicates the path coefficient intensity. * indicates significant difference at 0.05 level. ** indicates significant difference at 0.01 level. *** indicates significant difference at 0.001 level. GFI: goodness-of-fit index; CFI: comparative fit index; RMR: root mean square residual; RMSER: root mean square error of approximation.

## 4. Discussion

### 4.1. Effects of Different Restoration Measures on Soil

The fundamental chemical properties affected by the nature of salt include soil pH, EC, ES, and ESP [24]. In this study, all restoration treatments which applied FGD gypsum with humic acid led to a decrease in soil pH, SAR, and ESP (see Figure 1). These findings align with those of previous research [8,25]. The application of FGD gypsum with humic acid increased the content of $K^+$, $Ca^{2+}$, $Mg^{2+}$, and $SO_4^{2-}$ and decreased the content of $Cl^-$ and $HCO_3^- + CO_3^{2-}$. The presence of $Ca^{2+}$ in FGD gypsum allows it to replace exchangeable $Na^+$ in soil colloids. Additionally, $Ca^{2+}$ can engage in a precipitation reaction with water-soluble $CO_3^{2-}$ and $HCO_3^-$ in saline–alkaline soil [8]. The inclusion of humic acid enhances the dissolution of gypsum [26,27], thereby augmenting the $Na^+$ replacement capacity. Although the dissolution rate of FGD gypsum is modest, the chemical reaction between $Ca^{2+}$ and water-soluble $CO_3^{2-}$ and $HCO_3^-$ is relatively swift. This results in a sharp decline in soil pH during the year of application, which corroborates our current findings.

In this study, the addition of FGD gypsum with humic acid led to an increase in soil salinity (see Figure 1b). The interaction effects of FGD gypsum with humic acid combined with biofertilizer increased the pH, soil salinity, ESP, $K^+$, $Na^+$, $Cl^-$, $HCO_3^- + CO_3^{2-}$, and ES. This outcome aligns with the findings reported by Zheng et al. [28] and Tian et al. [29]. We conducted a potted experiment without irrigation during the plant-growing period, resulting in the inadequate drainage of salts and an overall elevation in the total salinity of the potted soil. In this process, $Ca^{2+}$, acting as a salt, reacts with free $NaHCO_3$ and $Na_2CO_3$ in the soil, resulting in the formation of $CaCO_3$, $CaHCO_4$, and $Na_2SO_3$. While $Na_2SO_3$ can be leached through drenching, insufficient drenching can lead to an increase in the overall

soil salinity [7,28]. Despite the rise in total water-soluble salts caused by the application of an appropriate dose range of FGD gypsum, it did not negatively impact plant growth. On the contrary, reductions in soil salinity and alkalinity notably enhanced plant growth [7]. It needs more consideration about the impact of the mineral impurities in humic acid used for saline–alkali soil restoration. However, previous research in this domain has not yielded substantial information concerning the impurity of humic acid [30,31].

### 4.2. Effects of Different Restoration Measures on Plant Growth and Yield

In this study, the application of bio-fertilizer notably enhanced plant growth and biomass (see Figures 3 and 4). Under the conditions of bio-fertilizer application, the use of medium and low doses of FGD gypsum with humic acid proved to be more beneficial for increasing stem, root, and total biomass. This underscores that the combined application of FGD gypsum and bio-fertilizer is more effective, in line with the findings of Wang et al., 2015 [32]. The application of bio-fertilizers which formulated from *Bacillus subtilis*, *Bacillus licheniformis*, and *Brevibacillus breviscan* expedites the dissolution of desulphurization gypsum by enhancing soil structure [32], and the use of microbial mycorrhizal agents activates soil nutrients [33,34], intensifying the desalination effect on soil and fostering crop growth. The application of FGD gypsum leads to improvements in soil organic matter, soil physical properties, and soil microbial communities [35,36], in addition to promoting plant growth and enhancing plant stress tolerance [7]. Concurrently, the application of desulphurization gypsum bolsters soil structure and lowers the soil's pH value, thereby enhancing the effectiveness of bacterial fertilization [37,38]. Macromolecular humic substances exhibit the capacity to bind ions. These substances, rich in oxygen-containing acidic functional groups, possess robust ion exchange and complexation capabilities. They may also interact with salt separation agents, thereby diminishing the latter's effectiveness [39]. Humic acid further fosters the formation of soil aggregates, enhancing the aggregate structure of the soil, and regulating water, fertilizer, gas, and heat conditions in the soil, thereby enhancing the growth environment for crops. The addition of humic acid to desulfurized gypsum alleviates the detrimental effects of salt stress on plants. It reduces the binding rate of $Na^+$ to the plant cell wall membrane and improves the functioning of cell plasma membranes. This, in turn, enhances the salt tolerance and yield of plants [26,27]. Additionally, it promotes root growth by increasing the concentration of essential mineral nutrients in plants [40], along with the organic acids and residues secreted by their roots. The organic acids secreted by roots, as well as those produced by microbial decomposition, also neutralize soil alkalinity. Moreover, the return of litter (roots, stems, leaves) to the soil enhances soil structure, augments soil organic matter, and boosts soil fertility [41].

### 4.3. Effects of Different Restoration Measures on Plant Ion Content and Transport

We observed that bio-fertilizer significantly increased the content of $Mg^{2+}$ in the leaf (see Figure 5) and the ratios of $Ca^{2+}/Na^+$ and $Mg^{2+}/Na^+$ in the leaf (see Figure 6). It also facilitated the transport of $Ca^{2+}$ and $Mg^{2+}$ from stems to leaves (see Figure 7). The application of FGD gypsum with humic acid significantly raised the leaf content of $K^+$ (see Figure 5) and the ratios of $K^+/Na^+$ and $Mg^{2+}/Na^+$ in leaves (see Figure 6), and it promoted the transport of $Mg^{2+}$ from stems to leaves (see Figure 7). Overall, these findings suggest that the selective uptake of $K^+$, $Ca^{2+}$, and $Mg^{2+}$ by *M. sativa* leaves significantly increased due to the restoration measures, as did the transport of beneficial ions from stems to leaves. This coordinated physiological response should alleviate the ion-toxic effects of saline–alkali soils on plants, ensuring the normal functioning of their leaves [12].

Salinity and alkalinity stress primarily affect plants by disrupting osmosis, leading to physiological drought, ionic poisoning of tissues and cells, and hindering nutrient uptake [3]. Maintaining the ionic equilibrium within plant cells is crucial for stabilizing the intracellular environment. However, adverse abiotic conditions such as high temperature, salinity, and frost damage can disrupt this balance, impairing normal metabolic processes [42]. Elevated levels of Na in the soil solution can hinder the K nutrition of

plants [43]. Therefore, it is recommended to maintain an optimal K level in salt-affected soils for optimal plant growth [44], development, and yield. In this context, $K^+$ plays a critical role as an osmoregulatory ion. A higher concentration of $K^+$ can mitigate the osmotic stress effect in saline soil, enhancing the salt tolerance of plants. $Ca^{2+}$ acts as an essential signaling molecule, contributing to cellular stability and safeguarding cell membrane structure under salt stress without affecting $K^+$ quality in plants. This helps alleviate the effects of salt stress and even enhances the selective uptake and transport of $K^+$ to reinforce the ionic balance [17]. Another cation, $Mg^{2+}$, proves advantageous in enhancing photosynthesis in leaves, improving light energy utilization, and meeting the light energy requirements for plant growth, thereby boosting salt tolerance. Upon the absorption of $Na^+$ from saline soils with high $Na^+$ concentrations and subsequent accumulation in saline environments, plants face reduced ability to absorb K, P, Ca, and other nutrients due to the competition for $Na^+$ [8,45]. By regulating ion distribution and transport processes in plants, effective restoration measures for saline–alkali land can mitigate osmotic stress and ion toxicity. This, in turn, promotes plant growth and development and enhances salinity tolerance. When applied to saline sites, the $Ca^{2+}$-rich FGD gypsum supports the "potassium enrichment and sodium rejection" process in plants via $Ca^{2+}$ aggregation [46]. Our research aligns with the findings of the aforementioned literature. Comparatively, the selective transport ratios of beneficial ions, namely $K^+$, $Ca^{2+}$, and $Mg^{2+}$, from stems to leaves were generally higher post-restoration. This suggests that the experimental measures bolstered the internal transport of beneficial ions within plants. This, in turn, reduces the damage incurred by excess $Na^+$ exposure and input to plants.

In this study, we observed a strong correlation between plant growth and the content and transport of $Mg^{2+}$ within the plant body (see Figure 9). The $Mg^{2+}/Na^+$ ratio of leaf and the selective transport ratios of $Mg^{2+}$ from stems to leaves showed varying degrees of increase under different levels of biofertilizer and FGD doses in each level of biofertilizer (see Figures 6 and 7). Furthermore, the selective transport ratios of $Mg^{2+}$ from stems to leaves increased under the interaction effect of biofertilizer combined with FGD and humic acid. This indicates that these interventions enhanced the plant's ability to selectively absorb $Mg^{2+}$ in its leaves, ultimately improving salt tolerance and promoting healthy growth. $Mg^{2+}$ plays a crucial role in numerous physiological and biochemical processes during plant development and growth. Approximately 35% of atmospheric $Mg^{2+}$ is transported to chloroplasts for photosynthesis. Beyond its role in light reactions as a component of chlorophyll, $Mg^{2+}$ also activates photosynthetic enzymes for carbon fixation [47–49]. Moreover, $Mg^{2+}$ acts as an activator for many enzymes in plants, and a deficiency in Mg can reduce the efficiency of carbon assimilation, subsequently lowering photosynthetic efficiency [50]. Additionally, $Mg^{2+}$ supports protein synthesis and nitrogen metabolism, both of which are integral processes for plant growth. In particular, $Mg^{2+}$ influences nitrogen metabolism by regulating the activity of key enzymes such as nitrate reductase [51].

### 4.4. Restoration Measure–Soil–Plant Correlations

Our results revealed a close correlation between the content and transport of $Ca^{2+}$, $Mg^{2+}$, and, to a lesser extent, $K^+$ within the body of *M. sativa* and its growth in response to the restoration measures (see Figures 7 and 8). The modification of FGD gypsum with humic acid and bio-fertilizer notably increased the transport of these beneficial ions (see Figure 9), with the addition of bio-fertilizer significantly amplifying this effect.

In this study, soluble $K^+$, $Mg^{2+}$, and $HCO_3^-$ in the soil had notable effects on root growth, with soluble $K^+$ exhibiting the most pronounced impact (see Figure 8). The application of FGD gypsum with humic acid significantly increased the amounts of $K^+$, $Mg^{2+}$, and $Ca^{2+}$ ions in the soil. This facilitated the root uptake of these beneficial ions, thereby safeguarding their normal transport within the *M. sativa* plant and supporting their corresponding physiological functions (see Figure 9). Both $Mg^{2+}$ and $Cl^-$ in the soil play a role in promoting plant growth by facilitating the selective uptake of $Ca^{2+}$ and $Mg^{2+}$ by organs and their subsequent transport to leaves.

## 5. Conclusions

Overall, our results suggested that the application of FGD gypsum and humic acid can decrease soil alkalinity and increase beneficial ions' amounts in the soil. Bio-fertilizers strongly promoted the growth and biomass of plants, ultimately enhancing the translocation of key ionic components to leaves. A combination of biofertilizer, FGD gypsum, and humic acid has the potential to increase the biomass, enhancing the translocation of $Mg^{2+}$ to leaves (see Figures 6 and 7). In terms of *M. sativa*'s biomass, the most beneficial treatment combinations were found to be 15.0 $g \cdot kg^{-1}$ of FGD gypsum + 1.5 $g \cdot kg^{-1}$ of humic acid as a package combined with 6.0 $g \cdot kg^{-1}$ of bio-fertilizer (see Figures 3 and 4), which increased the plant biomass for 386.81%, or 7.50 $g \cdot kg^{-1}$ of FGD gypsum + 0.75 $g \cdot kg^{-1}$ of humic acid as a package combined with 6.0 $g \cdot kg^{-1}$ of bio-fertilizer, which increased the plant biomass for 313.44% (see Figure 4). Our results offer new insights into the interactions among restoration strategy, soil, and plants in saline–alkali land restoration, providing practical solutions for the restoration of saline–alkali soil. Research in the future will be required to understand the regulation mechanisms between above- and belowground parts as well as their contributions to ion distribution, which, in turn, would be beneficial for achieving the aims of sustainable land restoration. Furthermore, research into how different soil amendments affect plant growth, particularly focusing on the impact of the impurities of various soil conditioners like humic acid.

**Author Contributions:** Data curation, B.Y.; Formal analysis, B.Y.; Funding acquisition, T.B.; Investigation, B.Y.; Methodology, T.B.; Project administration, T.B.; Resources, T.B.; Supervision, T.B.; Writing—original draft, B.Y.; Writing—review and editing, L.C. All authors have read and agreed to the published version of the manuscript.

**Funding:** This work was supported by the Research on key technologies of mixed grassland to adapt to climate change (2021GG0415) and 2020-Science and Technology Promote Development in Inner Mongolia-The Technological Innovation of Grass Seed Industry-2.

**Data Availability Statement:** The data presented in this study are available on request from the corresponding author.

**Acknowledgments:** We are grateful for the help of Fujin Zhang and Xiliang Li of the Institute of Grassland Research of the Chinese Academy of Agricultural Sciences (CAAS).

**Conflicts of Interest:** The authors declare no conflict of interest.

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
