# Peer review of "Effects of Restoration Strategies on the Ion Distribution and Transport Characteristics of Medicago sativa in Saline–Alkali Soil"

_agronomy, doi:10.3390/agronomy13123028_

Round 1
Reviewer 1 Report
Comments and Suggestions for Authors
The authors undertake interesting research issue on the combined effect of biofertilizer, gypsum and humic acids on the soil and the plant. I think that the above combination constitutes an element of novelty in agricultural research. The advantage of these studies is also their comprehensive nature, covering the impact of additives on both the soil and the plant. The mechanisms of binding and transport of micro- and macro-elements in soil and plants require in-depth research due to many ambiguous research results and conclusions, which in turn result from the multifactorial nature of the above processes. In my opinion, the work is interesting, but it should include some corrections.
Abstract: The abstract should be shortened. I recommend expressing significant changes as percentages to make it easier for the reader to understand the effects.
References should be given as numbers in the text of the manuscript. I think that authors should check and apply the instructions of the Agriculture journal.
“Measures” – this word is often used in the manuscript probably as “methods” and can introduce ambiguity to the text.
39-47: This part should be replaced by numeric information about saline-alkali lands. How many such soils are there? How quickly this problem is growing. Information from different years should be provided (or other relevant). The reader needs to see what the real problem is.
56: please rewrite the sentence. Humic acids are part of organic matter. “and” is not suitable here.
71-75: very complex sentence which has to be reduced.
75: which humic acid? From which materials extracted, In which form produced?
107: ESP, and SAR are not explained before first use.
113: Experimental site and Experimental design should be linked or can be separated but rewritten because in first- and second-chapter authors write about soil collection. However, in first one I do not have information about depth. It is in the second chapter.
119: 2961°C could you explain the sense of this?
132: Lack of information about the composition of humic acid is a serious mistake in this type of work. 25% of impurities may significantly affect the results! Perhaps the observed effect does not come from humic acids but from 25% of impurities! Please clarify this point. In addition, please describe the form of humic acid. I know from my experience that humic acids produced on an industrial scale are extracted with strong bases and then, for economic reasons, it is not purified from alkaline cations, which may cause it to have a high pH. Sometimes humic acid is neutralized with acid, but then the salinity of the product increases significantly. Please describe the properties of humic acid used in the research, as well as its composition and form. This is extremely important in this research.
134: which kind of bacteria?
134: please describe widely biofertilizer composition, form.
135: Table 1. Please provide properties of humic acids and biofertilizer
147: footnotes with explanation of the symbols have to be added to the table. Similarly, in the text, B0 and B6 as well as D0-D30 have to be explained.
158: provide reference or description for CEC
172: plant sample?
324: type of salt?
330: what’s about formation of insoluble calcium humates?
341: I think it depends on the kind of biofertilizer, microorganisms used for biofertilizers. Fungi can provide another effect that bacteria. And another effect can be for different fungi and different bacteria.
348-358: Why don't the authors also consider the impact of the 25% mineral impurities in humic acid?
455: If possible, please provide the most important increases or decreases as percentages. It will be easily understandable and accessible to the reader.
Author Response
|
Response to Reviewer X Comments
|
||||||||||||||||||||||||||||||||||||||||||||||||||||||||
|
1. Summary |
|
|
||||||||||||||||||||||||||||||||||||||||||||||||||||||
|
Thank you for your valuable comments for our manuscript. We would like to thank you for your time and effort. The suggestions are very important to improve our manuscript. According to your helpful advice, we have now incorporated all suggested changes into our manuscript and revised the manuscript carefully. Please find the detailed responses below and the corresponding revisions/corrections highlighted/in track changes in the re-submitted files. |
||||||||||||||||||||||||||||||||||||||||||||||||||||||||
|
2. Questions for General Evaluation |
Reviewer’s Evaluation |
Response and Revisions |
||||||||||||||||||||||||||||||||||||||||||||||||||||||
|
Does the introduction provide sufficient background and include all relevant references? |
Yes/Can be improved/Must be improved/Not applicable |
|
||||||||||||||||||||||||||||||||||||||||||||||||||||||
|
Are all the cited references relevant to the research? |
Yes/Can be improved/Must be improved/Not applicable |
|
||||||||||||||||||||||||||||||||||||||||||||||||||||||
|
Is the research design appropriate? |
Yes/Can be improved/Must be improved/Not applicable |
|
||||||||||||||||||||||||||||||||||||||||||||||||||||||
|
Are the methods adequately described? |
Yes/Can be improved/Must be improved/Not applicable |
|
||||||||||||||||||||||||||||||||||||||||||||||||||||||
|
Are the results clearly presented? |
Yes/Can be improved/Must be improved/Not applicable |
|
||||||||||||||||||||||||||||||||||||||||||||||||||||||
|
Are the conclusions supported by the results? |
Yes/Can be improved/Must be improved/Not applicable |
|
||||||||||||||||||||||||||||||||||||||||||||||||||||||
|
3. Point-by-point response to Comments and Suggestions for Authors |
||||||||||||||||||||||||||||||||||||||||||||||||||||||||
|
Comments 1: The abstract should be shortened. I recommend expressing significant changes as percentages to make it easier for the reader to understand the effects. |
||||||||||||||||||||||||||||||||||||||||||||||||||||||||
|
Response 1: Thank you for pointing this out. I agree with this comment. I have amended the abstract accordingly as advised.
|
||||||||||||||||||||||||||||||||||||||||||||||||||||||||
|
Comments 2: References should be given as numbers in the text of the manuscript. I think that authors should check and apply the instructions of the Agriculture journal. |
||||||||||||||||||||||||||||||||||||||||||||||||||||||||
|
Response 2: Thank you for pointing this out. I agree with this comment. I referred to Agricultural journal instructions and amended the abstract accordingly as advised. |
||||||||||||||||||||||||||||||||||||||||||||||||||||||||
|
Comments 3: “Measures” – this word is often used in the manuscript probably as “methods” and can introduce ambiguity to the text. Response 3: Thank you for pointing this out. I agree with this comment. I have amended as advised.
Comments 4: 39-47: This part should be replaced by numeric information about saline-alkali lands. How many such soils are there? How quickly this problem is growing. Information from different years should be provided (or other relevant). The reader needs to see what the real problem is. Response 4: Line 41-47:
Comments 5: please rewrite the sentence. Humic acids are part of organic matter. “and” is not suitable here. Response 5: Thank you for pointing this out. I agree with this comment. I have amended this sentence as advised. Line 60: Combining FGD gypsum with other soil-conditioning materials such as organic fertilizer and humic acid can decrease soil pH,
Comments 6: 71-75: very complex sentence which has to be reduced. Response 6: Thank you for pointing this out. I have amended this sentence into 3 sentences. Line 75-79:
Comments 7: which humic acid? From which materials extracted, In which form produced? Response 7: Thank you for pointing this out. We added properties of humic acid as below. Table 2. Properties of the humic acid used in the research.
Comments 8: 107: ESP, and SAR are not explained before first use. Response 8: Thank you for pointing this out. I agree with the comment. To make it more specific, I added explanations for two specific terms as shown below. Line 110-111: restoration measures can variably reduce soil pH, exchangeable sodium percentage (ESP), and sodium adsorption ratio (SAR)…
Comments 9: 113: Experimental site and Experimental design should be linked or can be separated but rewritten because in first- and second-chapter authors write about soil collection. However, in first one I do not have information about depth. It is in the second chapter. Response 9: Thank you for pointing this out. I agree with this comment. I have amended as advised.
Comments 10: 119: 2961°C could you explain the sense of this? Response 10: Thank you for pointing this out. I agree with the comment. 2961 ℃ mentioned in the article refers to the active accumulated temperature. It means the sum of daily active temperatures during a certain period or a certain growing season of crops. It is an important indicator of the heat resources of a place and the heat requirements for crop growth and development. Active accumulated temperature is widely used in agricultural climate analysis, agricultural climate zoning and agricultural meteorological forecast. Generally, the entire period during which the daily average temperature is maintained stably at ≥10°C throughout the year is the active period for the growth of various plants. The sum of the daily average temperatures during this period is called active accumulated temperature. Due to article limitations, we have not provided explanations for its definition.
Comments 12: 132: Lack of information about the composition of humic acid is a serious mistake in this type of work. 25% of impurities may significantly affect the results! Perhaps the observed effect does not come from humic acids but from 25% of impurities! Please clarify this point. In addition, please describe the form of humic acid. I know from my experience that humic acids produced on an industrial scale are extracted with strong bases and then, for economic reasons, it is not purified from alkaline cations, which may cause it to have a high pH. Sometimes humic acid is neutralized with acid, but then the salinity of the product increases significantly. Please describe the properties of humic acid used in the research, as well as its composition and form. This is extremely important in this research. Response 12: Deeply appreciate your invaluable insight. I strongly agree with your viewpoint. I have added properties of humic acid as below, wish it will helpful. Table 2. Properties of the humic acid used in the research.
Comments 13: 134: which kind of bacteria? Response 13: Thank you for pointing this out. I agree with this comment. I have specified as advised. Line 138-139: the bio-fertilizer was produced by Shandong Jinyao Biotechnology Co., Ltd., the composition of the biofertilizers is Bacillus subtilis, with an effective live bacterial count of ≥200 × 108·g−1.
Comments 14: 134: please describe widely biofertilizer composition, form? Response 14: Thank you for pointing this out. I agree with this comment. I have specified as advised. Line 138-139: the bio-fertilizer was produced by Shandong Jinyao Biotechnology Co., Ltd., the composition of the biofertilizers is Bacillus subtilis, with an effective live bacterial count of ≥200 × 108·g−1.
Comments 15: 135: Table 1. Please provide properties of humic acids and biofertilizer Response 15: Thank you for pointing this out. I agree with this comment. I have specified as advised. Table 2. Properties of the humic acid used in the research.
Comments 16: 147: footnotes with explanation of the symbols have to be added to the table. Similarly, in the text, B0 and B6 as well as D0-D30 have to be explained. Response 16: Thank you for pointing this out. I agree with this comment. I have added the footnotes as advised and explained them in the text as well. Line 147-150: The field experiment utilized a two-factor split-plot design, with the main plot receiving the applied bio-fertilizer (B) factor at two levels, which are 0 g·kg-1 (B0) and 6.0 g·kg-1(B6). While the subplot received FGD gypsum with humic acid (D) at four levels, which are 0 g·kg-1 (D0), 7.5+0.75 g·kg-1 (D7.5), 15.0+1.5 g·kg-1 (D15), 30.0+3.0 g·kg-1 (D30). Line 159-161: Note: B0 and B6 indicates the treatments applied Bio-fertilizer with 0 g·kg-1 and 6.0 g·kg-1. D0, D7.5, D15, D30 indicates the treatments applied FGD gypsum+Humic acid with 0 g·kg-1, 7.5+0.75 g·kg-1, 15.0+1.5 g·kg-1, 30.0+3.0 g·kg-1, respectively.
Comments 17: 158: provide reference or description for CEC Response 17: Thank you for pointing this out. I agree with this comment. I have amended as advised. Line 172-174:
Comments 18: 172: plant sample? Response 18: Thank you for your valuable comments. I have rewritten this part to make it more specific. Line 188-191:
Comments 19: 324: type of salt? Response 19: Thank you for pointing this out. I agree with this comment. I have rewritten this sentence. Line 87-399:
Comments 20: 330: what’s about formation of insoluble calcium humates? Response 20: Thank you for your valuable comments.
Comments 21: 341: I think it depends on the kind of biofertilizer, microorganisms used for biofertilizers. Fungi can provide another effect that bacteria. And another effect can be for different fungi and different bacteria. Response 21: Deeply appreciate your invaluable insight. I agree with this comment. Line 404-407:
Comments 22: 348-358: Why don't the authors also consider the impact of the 25% mineral impurities in humic acid? Response 22: Thank you for pointing this out. I agree with this comment. I have added some consideration about the impact of impurities of humic acid in discussion section as advised.
Comments 23: 455: If possible, please provide the most important increases or decreases as percentages. It will be easily understandable and accessible to the reader. Response 23: Thank you for such helpful suggestion. I have revised the related parts of article as advised.
4. Response to Comments on the Quality of English Language |
||||||||||||||||||||||||||||||||||||||||||||||||||||||||
|
Point 1: |
||||||||||||||||||||||||||||||||||||||||||||||||||||||||
|
|
||||||||||||||||||||||||||||||||||||||||||||||||||||||||
|
5. Additional clarifications |
||||||||||||||||||||||||||||||||||||||||||||||||||||||||
|
We tried our best to improve the manuscript and made some changes in the manuscript. These changes will not influence the content and framework of the paper. We appreciate for Editors/Reviewers’ warm work earnestly, and hope that the correction will meet with approval. Once again, thank you very much for your comments and suggestions. |
||||||||||||||||||||||||||||||||||||||||||||||||||||||||

Reviewer 2 Report
Comments and Suggestions for Authors
This article presents a very broad experimental review the distribution and transport dynamics of cations in plants is crucial for understanding their response mechanisms to saline-alkali stress conditions.
Within the context of the work, it can be mentioned that the topic is interesting because it provides a new insights into the interactions among measures, soil, and plants in saline-alkali land restoration, providing practical solutions for the restoration of saline-alkali soil.
I found it interesting that the search for information was done through an experimental design, a methodology that is very useful when it is required to verify different experimental parameters and their effect within an analytical process.
The introduction presents sufficient preliminary information to understand the foundation and purpose of the scientific work developed by the authors. Likewise, an evaluation was made of probable plagiarism, but nothing was found that indicated it, so it is considered that this is a completely original work.
It is worth mentioning that the experiments within the experimental design were adequately directed, the information presented on the experimental methodologies carried out is perfectly founded both in its proposal within the experimental determinations, as well as in the particular application.
The graphs perfectly show the results obtained.
The statistical tools used show consistent results within the analysis carried out by the authors.
It is considered that the article has the necessary qualities to be published
While the authors should address some issues before acceptance
· Line 107: What are ESP and SAR??
· Line 119: 2961 °C, Please revise.
· Line 133: the authors should provide some details about the composition of the biofertilizers used in this experiment.
· Lines 162-170: provide a reference
· In the Figures: Please number the figures from a ..to …z and mention that in the figure captions.
· In the discussion section: The authors should Tables and Figs to be in contact with the results.
· In the conclusion section: Instead of repeating the results here, what can be done after this study or suggestions should be made. What is the novelty of this work? Your study should conclude if the obtained results can help farmers?? What are the future perspectives????
Comments on the Quality of English LanguageMinor editing of English language required
Reviewer 3 Report
Comments and Suggestions for Authors
Presentation of the analysis results needs to be revised using the statistical procedures for means comparison, especially for those measurement variables showing a significant interaction. A significant effect of treatment combinations does not automatically means significant interaction, so comparison between treatment combinations does not the way to show interaction comparison.
In this case, B6D30 cannot be compared with B0D15 (Fig.1). If for example the are significantly different, are they different due to biofertilizer or higher doses? Nobody knows unless further experiment done for that.
Statistically, a significant interaction effect means significantly different response to the second factor between each level of the first factor. Example in this study, biomass difference between B0 & B6 may not significant in D0 (no effect of biofertilizer), but as the doses increased the differences may be significant, or vise versa, under B0 the effect of doses may not significant but it is significant under B6.
I have given some example of inappropriate comparison in each table to highlight that the data analysis and presentation of the results need to be revised.

Round 2
Reviewer 1 Report
Comments and Suggestions for Authors
Table 2: Why the content of humic acids is higher than content of organic matter? Humic acids are part of organic matter.
Reviewer 3 Report
Comments and Suggestions for Authors
Comments of the figures (Fig.1 to Fig.7) for presentation of the analysis results: In a two factor experiment, say factors B & D, then the authors are interested in testing a hypothesis whether the effect of B is significant (i.e. there is a significant deference between levels of B, for example between B0 & B1) and other hypothesis whether the effect of D is significant (i.e. there is a significant difference between levels of D, for example between D0 & D1 or D0 & D2 or D0 & D3, etc). If the interaction is significant on a specific measurement variable, say total biomass, it means that the total biomass responses to the factor D are sigificantly different between levels of B and those responses to the factor B (say to biofertilizer) are significantly different between levels of D. This could mean that under B0 there are no significant differences between levels of D but under B1 there is at least one pair of D is significantly different, or vice versa, for example under D0 there is no significant different between B0 & B1 but under other levels of D, there is at least one level of D showing significant different between B0 & B1.
In this experiment, the author cannot compare between treatment combinations to show or to interprete the significance of an interaction effect because not all treatment combinations are comparable, for example comparing between B0D0 & B1D1 is bias because if they are significantly difference then the authors and nobody can answer whether they are different due to increased B level or D level, so that is a biased comparison (see my previous comments).
The facts in this version 2 of the paper, the presentation of the analysis results are incorrect and confusing. Let see Fig.2 (e) for example, the horizontal lines are misleading and do not the correct results of data analysis. According to the Statistics convention, mean values sharig the same letters are to show non-significant deferences of multiple comparison between them. The horizontal line with the letter "a" points to mean values of B0D7.5 in the left and B6D7.5 in the right but it is not defined what "a" means. In fact, these two mean values are significantly different (based on the mean & SE bar). The correct way to show that they are significantly different is by assigning the B6D7.5 bar with "A" and B0D7.5 bar with "B". The next horizontal line has the letter "b", but what doeas it means. In fact, the pointed mean values are not significantly different based on their mean values & SE bars. The next two horizontal lines shares the same letter "c", but what do they mean? Are they showing non-significant difference? In fact, the upper "c" line points to two mean values that are significantly different (B6D15 > B0D15), while the lower "c" line points to two mean values that are not significantly different (comparison between B0D30 & B6D30 is non significantly different).
Please read my full comments in the attached file.

I think the English is fine, but in some sentences, there are some inappropriate sentence structures, such as incomplete sentence, and too long sentence with poor structure, which make the meaning become not clear.
